# Micro-optical Components for Bioimaging on Tissues, Cells and Subcellular Structures

**DOI:** 10.3390/mi10060405

**Published:** 2019-06-19

**Authors:** Hui Yang, Yi Zhang, Sihui Chen, Rui Hao

**Affiliations:** 1Laboratory of Biomedical Microsystems and Nano Devices, Bionic Sensing and Intelligence Center, Institute of Biomedical and Health Engineering, Shenzhen Institutes of Advanced Technology, Chinese Academy of Sciences, Shenzhen 518055, China; sh.chen@siat.ac.cn (S.C.); rui.hao@siat.ac.cn (R.H.); 2Institute of Biomedical Therapeutics, University of Southern California, Los Angeles, CA 90033, USA; zhangelliot@gmail.com; 3Pen-Tung Sah Institute of Micro-Nano Science and Technology, Xiamen University, Xiamen 361005, China

**Keywords:** micro-optics, bioimaging, microtechnology, microelectromechanical systems (MEMS), in vitro, in vivo

## Abstract

Bioimaging generally indicates imaging techniques that acquire biological information from living forms. Among different imaging techniques, optical microscopy plays a predominant role in observing tissues, cells and biomolecules. Along with the fast development of microtechnology, developing miniaturized and integrated optical imaging systems has become essential to provide new imaging solutions for point-of-care applications. In this review, we will introduce the basic micro-optical components and their fabrication technologies first, and further emphasize the development of integrated optical systems for in vitro and in vivo bioimaging, respectively. We will conclude by giving our perspectives on micro-optical components for bioimaging applications in the near future.

## 1. Introduction

Nowadays, bioimaging has enabled us to dig out biological information from deep inside of our bodies, and also revolutionized the way we understand, detect, and treat diseases in different angles and dimensions. In the past couple of decades, the thriving of digital computing has paved the way for a wide variety of imaging techniques, including ultrasound, computed tomography (CT), magnetic resonance imaging (MRI), etc. [1]. Through different mediums other than light, these imaging modalities can harvest spatiotemporal parameters from living organisms such as concentration, tissue functionality, anatomical morphology. Modern clinics utilize these different imaging modalities to acquire metabolic and anatomical information from a patient. Against lung tumors and bone metastasis, CT has been extensively used given its short imaging time and high spatial resolution. The co-registration of microCT imaging and volumetric decomposition has proven valuable to study cell trafficking, tumor growth, trabecular bone microarchitecture, and response to therapy in vivo [2]. However, it is not often used in soft tissue scans since the X-ray absorption is rather low in soft tissue, fat, neurons, hence yielding low resolution and inaccuracy in diagnosis [3]. As a complementary method, MRI has its unique advantages in monitoring soft tissue abnormality, brain and neural activity. Specifically, different research objects are most well-fitted correspond to distinctive modalities. By functional MRI (fMRI), the metabolism of neural function can be revealed by mapping the contrast variation in blood flow in response to specific stimulus [4]. Magnetic resonance spectroscopy (MRS) has the ability to identify various biochemical markers of neoplasm in isolated voxels which are three-dimensional pixels, and therefore has been successfully employed in regard to brain, breast, and prostate cancer [5,6,7]. While the hazards of radiation and strong magnetic field are now well-controlled in most medical contexts, ultrasound imaging proposes a substitute with low health risks. Using acoustic pulses to reflect the contrast between different tissues and objects, ultrasound imaging has had a tremendous impact in hemodynamics and inflammatory study in the past decade [8]. Recent novel use of microbubbles or nanoparticles as indicators has allowed ultrasound to monitor and regulate on molecular level [9,10].

Besides these imaging techniques, as researchers further acquire enhanced resolution in both clinical and experimental imaging applications, optical microscopy plays a predominant role in observing tissues, cells and biomolecules, as this visualization technique has been able to literally and figuratively illuminate the inner workings of cells, for example by using fluorescent probes to light up proteins and subcellular structures. Leveraging the characteristic emissions of biological fluorophores, optical imaging technique is possible to gain insights on cell structures and functions [11]. Moreover, since the emission diffraction barrier, i.e., the conventional resolution limit, can be overcome by stimulated emission depletion (STED) microscopy etc., researchers can now image fluorophore-labelled systems with a resolution of 15 nm, i.e., 3000 times smaller than the width of a single human hair [12,13,14].

Along with the fast development of microtechnology, there is an intensive demand on developing miniaturized and integrated optical imaging systems. Over the last decade, a fast-growing interest was notice for developing microdevices that integrate one or several optical functionalities/components onto a single chip with size being of only millimeter-size up to a few square centimeters in size [15]. These optical microdevices have shown great potential on imaging living organisms, tissues, cells, as well as subcellular structures, both in vivo and in vitro. The in vivo imaging is usually achieved by integrating such microdevices into imaging instruments, e.g., endoscopy and optical coherence tomography (OCT) instruments. While the in vitro imaging can be performed by using microfluidic devices, the latter can provide prominent advantages on sample pre-treatment and handling. In these review, we focus on recent developments in the realization and use of micro-optical components for bioimaging applications. Typical micro-optical components used in bioimaging instruments are introduced at first, enabling us to sketch a blueprint of today’s integrated imaging systems. Different technologies for the fabrication of micro-optical components are demonstrated. We further present the integration of these micro-optical components with instruments or devices for in vitro and in vivo bioimaging, respectively. We will conclude by giving our perspective on bioimaging with micro-optical components, especially on how this technology will impact modern biological/clinical study. We hope the review provides the reader with some orientation in the field and enables selecting platforms with appropriate characteristics for his/her application-specific requirement.

## 2. Micro-optical Components

Based on basic optical principles, one can classify micro-optical components into (i) refractive optical components that rely on the change of the refractive index at an interface, such as lenses, prisms and mirrors, (ii) diffractive optical structures that enable shaping of an optical beam by diffractive/interference effects, such as diffraction gratings, and (iii) hybrid (refractive/diffractive) structures. Refractive and diffractive optical components share many similarities when they are used to manipulate monochromatic light but their response to broadband light is very different. For a material with normal dispersion, refractive lenses have larger focal distances for red light than for blue light and prisms deflect longer wavelengths by a smaller angle; the contrary occurs for diffractive lenses and gratings. This contrasting behavior arises because two different principles are used to shape the light: refractive optics relies on the phase that is gradually accumulated through propagation, while diffractive optics operates by means of interference of light transmitted through an amplitude or phase mask. The decision to use diffractive or refractive optics for a specific optical problem depends on many parameters, e.g., the spectrum of the light source, the aimed optical application (beam shaping, imaging, etc.), the efficiency required, the acceptable straylight, etc. Arbitrary wavefronts can be generated very accurately by diffractive optics. A drawback for many applications is the strong wavelength-dependence. Diffractive optics is therefore mostly used with laser light and for non-conventional imaging tasks, like beam shaping, diffusers, filters and detectors. Refractive optical elements have in general higher efficiency and less stray light, even though in some cases it is more difficult to make refractive lenses with precise focal lengths or aspheric shapes. Moreover, for broadband applications, diffractive optical elements (DOEs) can be combined with refractive optics to correct for the chromatic aberration. This combination allows systems with low weight or which consist of only one material. In this section, we introduce in the following the most commonly used micro-optical components in bioimaging systems, categorized by their functionalities, including waveguides, mirrors and lenses.

### 2.1. Waveguides

An optical waveguide is a physical structure that transmits light along its axis, which is generally composed of a core with a cladding part. A planar optical waveguide is fabricated in a flat format and is particularly interesting to integrate into an imaging system. Recent research works have shown the great potential of optical waveguides in photonic integrated circuits based on high refractive index contrast (HIC) between the core and the cladding. Spiral waveguide geometries in HIC waveguides can be used to significantly increase the interaction length between the sample and the evanescent field of the waveguide [16], opening opportunities to develop, e.g., chip-based nanoscopy [17] and on-chip OCT [18]. Due to its suitable material properties and the compatibility of its fabrication process with standard complementary metal–oxide–semiconductor (CMOS) fabrication line, silicon nitride (Si_3_N_4_) has attracted the maximum attention. The suitable material property of Si_3_N_4_ includes transparency in visible wavelength, low absorption and high refractive index contrast to the cladding layer (typically SiO_2_). Being transparent with low auto-fluorescence and low absorption in the visible range makes Si_3_N_4_ compatible with fluorescence techniques for bioimaging.

Tinguely et al. presented the usage of a Si_3_N_4_ waveguide platform for integrated optical microscopy for in vitro bioimaging applications [19]. A SiO_2_ layer was first grown thermally on a silicon chip, followed by the deposition of Si_3_N_4_ layer using low-pressure chemical vapor deposition (LPCVD). Standard photolithography was employed to define the waveguide geometry using photoresist, and reactive ion etching (RIE) used to fabricate a waveguide rib of given height. The remaining photoresist was removed, and finally a top cladding layer was deposited by plasma-enhanced chemical vapor deposition (LPCVD). This low-loss Si_3_N_4_ waveguide platform was used to set up an evanescent field for total internal reflection fluorescence (TIRF) microscopy. The sample placed directly on top of the chip was illuminated by the evanescent field of the optical waveguide (Figure 1). In such waveguide chip-based microscopy, the illumination and collection light paths can be efficiently decoupled, opening several opportunities for bioimaging, e.g., on living cells [19] and on single molecules with super-resolution capability [17]. Besides, as the evanescent field is generated along the entire length of the waveguide, a low magnification objective lens can be employed to acquire TIRF images over a large field-of-view of even millimeter range. Moreover, as the evanescent field decays exponentially at the interface between the biological sample and the waveguide, only a thin, typically 100–200 nm section away from the surface can be illuminated, providing a high signal-to-noise ratio by reducing the background signal. 

### 2.2. Mirrors

The effectiveness of microelectromechanical systems (MEMS) in biomedical imaging has been demonstrated in many research findings, where micromirrors used for beam deflection and shaping are one of the core components [20,21]. Commercially available deformable micromirrors that employ MEMS technology are now a common method of reducing astigmatisms and aberrations, increasing the resolution of the imaging system [22]. Micromirrors capable of dynamic focus as well as two-dimensional (2D) scanning have been fully integrated [21]. Such devices rely on electrostatic actuation for both focus and beam deflection. Imaging systems integrated with micromirrors can either work by scanning the micromirror in the form of a raster, or by using a Lissajous scan format. The raster scan uses a fast axis and perpendicular slow axis simultaneously to form a uniform projection area, while the Lissajous scan is performed by exciting a bi-directional mirror at resonance along two perpendicular axes. Comparing this two techniques, the Lissajous scan method is more prominent in imaging applications as such method can provide high-resolution images, but this method also requires more computational power [23]. 

As an example, Morrison et al. presented a MEMS micromirror using electrothermal actuation [24]. In this work, the fabrication process included three highly doped polysilicon layers, two sacrificial oxide layers, and a gold layer patterned using optical lithography. Residual compressive stresses in the polysilicon layer that were combined with residual tensile stresses in the gold layer due to the fabrication process provided a stress gradient along the boundary of the gold and polysilicon layers. Upon release, an initial curvature can be generated because of a bending strain due to the stress in the bimorph structures. The difference in coefficient of thermal expansion of the two layers can provide actuation, providing a temperature dependent curvature (Figure 2). Janak et al. proposed developing and integrating a three-dimensional (3D) micromirror for large deflection scanning in in vivo OCT [25]. A two-axes scanning micromirror was fabricated by using high-aspect-ratio deep reactive ion etching (DRIE) process instead of anisotropic etching of silicon in aqueous solution of potassium hydroxide (KOH), highly reducing the dead space on the chip and achieving a high-degree of integration. The micromirror was used to steer the scattered light from the tissue, and the steered signal was combined with a reference light beam at an optical coupler to produce interference patterns, which were collected at a detector to produce 2D cross-sectional image of the tissue structures.

### 2.3. Lenses

The rapid growth of micro-opto-electro-mechanical systems (MOEMS or Optical MEMS) has attracted great interest in the field of microchip-based biophotonics for bio-sensing and high-resolution bioimaging [26]. One of the most important components in many optical micro-devices is a microlens, which can be integrated with emitters and detectors to improve the optical efficiency. In order to obtain high quality images for the application of some related fields in bio- and opto-electronics, high light focusing efficiency or high numerical aperture (NA) of the microlenses should be achieved. These two parameters are related to the geometry, particularly the curvature, of the microlenses. To date, many different fabrication technologies have been used for microlens fabrication, such as the photoresist reflow technique [27], photo- polymerization [28], LIGA (Lithographie, Galvanoformung, and Abformung, i.e., Lithography, Electroplating, and Molding) process [29], ink-jet printing [30], direct laser printing [31], and so on. 

Although these methods are able to fabricate microlenses, they have drawbacks that result from the multiple process steps that are quite complex and sophisticated. Therefore, it is necessary to introduce a more efficient technique that allows easy variation in terms of the microlens curvature in order to obtain high numerical apertures, which result in increasing image quality in bioimaging systems. It has been demonstrated that dielectric microspheres can be used as solid immersion microlenses to explore the possibility of super-resolution capability in recent years. Wang et al. used silica microspheres with diameter ~2–9 μm for super-resolution imaging in the far field by generating a magnified virtual image underneath the specimen [32]. Later, microspheres with high refractive index (e.g., barium titanate glass) have been also used in optical nanoscopy [33,34,35]. In these works, the microspheres were simply placed on top of the sample object, where they collected the underlying sample’s near-field nano-features and subsequently transformed the near-field evanescent waves into far-field propagating waves, creating a magnified image in the far-field, which is collected by a conventional optical microscope (shown in Figure 3). The super-resolution capability (imaging beyond the classical diffraction limit) of the microspheres, resulting from the enhanced optical field in the near field and the "photonic nanojet" phenomenon [35], has already been verified in bioimaging applications, such as molecular and subcellular structural characterizations [36,37].

## 3. Fabrication Technologies of Micro-optical Components

Fabrication of micro-optical devices mainly relied on techniques transferred from the conventional two-dimensional (2D) integrated circuit (IC) and two- or three-dimensional (3D) MEMS processes. Semiconductor fabrication has been adopted to create several types of structures on chips, including waveguides, photonic circuits, and lenses. This includes photolithography, thin film deposition, and chemical etching. Silicon-, glass-, glass-silicon-, glass-polymer-based fabrication techniques were widely studied. However, silicon and glass are hampered from wider applications in micro-optics, because they possess micromachining difficulties and are relatively expensive; moreover, an inconvenience of silicon is the lack of optical transparency at ultraviolet (UV), visible and near-infrared (IR) wavelengths. Tremendous effort has been made to find alternative materials that are more cost-effective and easier machinable. With the development of related fabrication techniques, polymer/plastic-based devices have therefore gained increasing interest. Compared with silicon and glasses, polymer materials can avoid high-temperature annealing and stringent cleaning (if they are disposable), they are more cost-effective, easier in microfabrication, and there exists a wider range of materials to be chosen for characteristics that are required for each specific application, such as good optical transparency, biocompatibility, and chemical or mechanical properties. However, polymer materials usually do not result in strongly bonded layers like glass or silicon, and can exhibit structural deformation during device packaging processes. Each material has therefore both its advantages and disadvantages, and the choice of it will depend on the specific application. New technologies have also been developed in the meanwhile. This section reviews current fabrication methodologies to make optical structures on a chip, focusing on integrated waveguides, micromirrors and microlenses.

### 3.1. Technologies for Waveguide Fabrication

Waveguides that are integrated on-chip can be categorized as based on two working principles, namely total internal reflection (TIR) and interference. TIR-based waveguides require that the refractive index of the core *n_c_* of the waveguide is bigger than that of the cladding *n_s_*. Interference-based waveguides are conceptually different. In these structures, light is multiple times reflected from a periodic dielectric cladding layer by wave interference, therefore, they do not require a cladding material with a lower index than that of the core material [15]. Integrated waveguides can be constructed using a variety of micromachining procedures. 

Common processes include lithographic patterning, thin-film deposition, and etching, these techniques can be used to fabricate a ridged waveguide, i.e., a solid-core waveguide. Usually, a thin layer of the core material is deposited on a planar substrate first. The substrate is coated with photoresist, the latter is then exposed to UV light or X-rays through a lithography mask that defines the waveguide shape, and developed to form a pattern on the surface of the substrate. With the remaining photoresist as a mask for either wet-chemical or dry etching to define the ridge waveguide structures. Dry etching methods, for example ion-beam etching, produce smooth edges, particularly on curved sections, but it also causes some lattice damage, which must be removed by annealing if minimum optical losses are desired. Wet chemical etching on the other hand produces less lattice damage, but it is very difficult to control the etch depth and profile. Most chemical etchants are preferential regarding crystal orientation, thus leading to ragged edges on curved sections of waveguides when using Si substrates. After the photolithography and etching procedures to process the core, a cladding layer is deposited. As an example, silicon oxynitride waveguides were fabricated in a standard silicon fabrication line by PECVD and LPCVD deposition processes in combination with 1100 °C annealing treatments to remove light-absorbing hydrogen bonds; and optical lithography for pattern definition and dry RIE for the pattern transfer process [38]. As shown in Figure 4, a 2 μm thick SiO_2_ buffer layer (n = 1.45) and a 1 μm thick SiO_x_N_y_ core layer (n = 1.85) were both deposited by LPCVD; a ∼ 0.5 μm thick borophosphosilicate glass (BPSG) layer (n = 1.45) was grown by LPCVD in order to optimize the surface planarity after RIE. Finally, the whole system was coated with a thin SiNx film (50 nm of thickness). This waveguide system was applied to detect low surface concentration (10^−11^ mol·cm^−2^) of a green light-emitting organic dye.

Focused-beam direct writing can also be used to define a solid-core waveguide, rather than using lithography and etching. Either electron beam or proton beam can work, the former was used to densify doped silica on a silicon substrate for fabricating a silica waveguide on a chip [39], and the latter was used to selectively slow down the rate of porous silicon formation during subsequent anodization [40]. When the doped silica was exposed to the electron beam, an appropriate change in the refractive index was generated, and the surface kept planar and was suitable for further integration, without the need for cladding layers [39]. The proton bombardment reduced the free-carrier density and increased the local resistivity of the material. During the subsequent electrochemical etching process, these defects acted as traps for holes and thereby avoided their migration to the silicon/electrolyte surface, reducing the rate of porous silicon formation in the exposed regions. This method generated a silicon core surrounded by a region of porous silicon that has a lower index of refraction [40]. Besides the solid-state waveguides, liquid medium can also be used as the core material of the waveguide. Many different formats of the liquid-core waveguide have been proposed, however, their applications in integrated bioimaging system are still limited and mainly stay in the research phase. 

### 3.2. Technologies for Micromirror Fabrication

MOEMS (micro-optical-electro-mechanical-systems) are one of promising techniques for developing optical switching connection devices [41]. A micromachined mirror is well known to be an optical key component. The advantages of using an optical micromirror include its low sensitivity to polarization and functionality in broadband optical applications. The micromirrors have been widely used in optical communication, for example, in optical scanners, projection display systems, variable optical attenuators, and other specific applications.

Prior works on micromirror fabrication have shown the feasibility of bulk or surface micromachining techniques, through silicon deep etching or using expensive silicon-on-insulator (SOI) wafers [42]. However, issues considered in the literature include high process complexity, high operation voltage or power consumption, etc. The standard CMOS process offers good opportunities for creating much smaller devices and more intelligent optical cross-connect devices with driving circuits on the same chip, based on a micromirror switch array, for optical telecommunication, steering light beams and communication applications. In such devices, the micromirrors are usually electrostatically actuated with a planar bottom drive that can steer a light beam in a continuous and controllable fashion [43]. Yoo et al. proposed an electrostatically actuated torsional micromirror [44]. The micromirror membrane was suspended by a pair of torsional springs, and an aluminum film, deposited by thermal evaporation, was used as a reflective material on the micromirror. A buried oxide (BOX) played the role of an insulating layer with a thickness of 1 μm and amorphous silicon was deposited on a glass substrate as the material for the electrodes, electrical lines and grounding shields. Figure 5 illustrates the two individually processed wafers of the micromirror, i.e., a bottom wafer with the addressing electrodes and a top wafer with the micromirror. Usually, the electrostatically actuated micromirror has a high fill factor, i.e., the ratio of the active reflecting micromirror area to the device area. Due to the high fill factor of the electrostatic micromirror, the discrete structure can be easily expanded to an array system without changing the basic device operating principles and its manufacturing process, such as the micromirror array based on the bending actuation of interdigitated cantilevers that was proposed by Kim et al. [45].

Besides the torsional micromirrors, a simple format with a 45° or an angular facet can be used as vertical optical interconnecting structure. These structures are relatively easy to fabricate, and their performances are generally acceptable. Performances of the devices using curved-shape mirrors are known better than the flat mirror devices, but curved mirrors are hard to fabricate. A variety of fabrication techniques have been utilized, such as excimer laser ablation, use of V-shaped 90° diamond blades to create a 45° cut, anisotropic etching of (100) silicon in potassium hydroxide solutions, modified reactive ion etch techniques relying on angular bombardment of ions, and so on [46]. Lee et al. proposed using silicon deep etch process and photoresist reflow process to fabricate a silicon structure which has curved structures that made of reflowed photoresist [47]. The fabricated structure can be used as a master structure for subsequent embossing process to produce a 12-channel waveguide device with curved-shape micromirror for vertical optical interconnection. Koh et al. presented a right-angle micromirror integrated into a polydimethylsiloxane (PDMS) microfluidic device for measurement of particles [48]. The micromirror with precise 45° reflection angle was fabricated using conventional microfabrication techniques including wet etch and soft lithography. A 90° V-groove was generated on a (100) silicon wafer by anisotropic etching first. After two steps of PDMS replication, commercial UV photopolymer was filled in the V-grooved PDMS channels and cured, followed by a metal layer deposition step on the hypotenuse of the polymer micromirror (as shown in Figure 6). 

### 3.3. Technologies for Microlens Fabrication

A microlens is an important key component for focusing and collimating light. It can be made up as a solid curved surface or as a tunable interface between two deformable materials/media. On-chip hard solid-state lenses usually are fixed-focus lenses, similar to miniaturized traditional lenses that are used in a free-space detection system. Different methods have been developed for building solid microlenses, including photoresist thermal reflow, direct writing, soft replica molding, ink-jet printing, etc.

Thermal reflow processes have been widely used for the fabrication of large arrays of hemispherical microlenses. In this technique, a glass substrate is first coated with a layer of photoresist. Photolithography technique is then used to pattern the photoresist layer and generate a series of cylindrical islands, which are heated above the glass transition temperature of the photoresist. Due to surface tension, the shape of the photoresist cylinders changes to minimize the surface energy and becomes hemispherical, therefore generating a microlens array on the substrate. As the surface tension governs the contact angle of the photoresist and the shape of the microlens eventually, a thin layer of additives can be deposited on the substrate to influence the surface tension between the substrate and the photoresist, tuning the focal length and numerical aperture of the fabricated lenses. Yang et al. utilized a bottom polyimide layer to form a pedestal to sustain the upper photoresist lens after the heat reflow process [49]. The interactive force between two material interfaces causes the upper photoresist to form a spherical profile and transform the polyimide pedestal into a trapezoid with arc sides (as shown in Figure 7a). Advantages of the thermal reflow technique include a low material consumption, low manufacturing costs, the intrinsic simplicity of the technique, and an easy process control. Microlens arrays can be produced over large surface areas. However, the low transparency and thermal instability of photoresist features during the reflow process limit its widespread application [15].

Another widely used technique to fabricate microlens is direct writing, including electron beam lithography [50], direct laser writing [51], focused ion beam writing [52], and laser ablation [53]. With the direct writing technique, the material is exposed and processed in a way that the local thickness of the material is a continuous and preferably linear function of the energy deposited by the beam. The major advantage of this technique is its potential to fabricate microlenses with characteristics that can be tailored for every lens individually. Typically, electron beam lithography and direct laser writing methods correlate the surface profile data of the desired microlens to the beam intensity values, the latter being synchronously modulated to write a continuous pattern in the material to generate a microlens (as shown in Figure 7b). Alternatively, focused ion beam writing and laser ablation methods rely on the principle of precision removal of substrate materials. The depth of cut is controlled by the dwell time at each step and the number of times that the area is scanned.

Besides, micromolding methods, such as compression molding, injection molding and soft replica molding, are most suitable for low-cost mass-production. Compression molding, including hot embossing, is a simple process with relatively low initial cost to replicate microlenses. The final height and radius of curvature of hot-embossed microlenses are determined by the combination of processing parameters, such as pressure, temperature and time. In the injection molding, sub-micron features in the mold are difficult to be filled because the solidified layer or solidified shell in a glassy state has very high viscosity. Therefore, the surface temperature of the mold can be maintained above the glass transition temperature of the polymer and then the micro/nanopatterns in the mold can be fully filled [54]. One can also use UV polymerization of a polymer in the mold. Soft replica molding based on soft lithography technology is widely used in molding PDMS and UV-curable materials and suitable to duplicate 3D topography in a single step. Convex and concave PDMS molds which define the microlens geometry can be fabricated first. Then both concave and convex microlenses can be fabricated by casting the polymeric liquid, for example polymethyl methacrylate (PMMA), in the corresponding convex and concave molds, respectively. 

## 4. Micro-optics Integrated within Bioimaging Systems

This section will review the main applications of micro-optical components for bioimaging on tissues, cells and subcellular structures. We will consecutively discuss in vitro and in vivo imaging applications.

### 4.1. Micro-optical Components for in vitro Bioimaging

Recent advances in photonics and imaging techniques offer new possibilities for optical imaging systems and sensor devices for fundamental studies and future biomedical applications. Ongoing developments suggest that an impressive diversity of biological and medical questions can be answered using optical microscopy. Advanced optical techniques have been proposed for achieving high-resolution imaging of sample of interest. For example, multi-photon microscopes can investigate cortical micro-architecture in animals’ brain with single cell resolution [55]. Also in regard to detecting levels of the tumor suppressing protein p53 in cells, fluorescence microscopy has been used to image cancerous tissues that were stained by specially designed oncolytic adenoviruses, the latter being programmed to replicate if the cellular p53 level is low [56,57]. However, these techniques are frequently accompanied by the need for high-end, bulky and expensive optical set-ups that are rather cumbersome to operate and to maintain, especially for untrained users. Such drawbacks have motivated researchers to minimize and integrate optical components in imaging systems for in situ analysis and observation of bio-samples under *in vitro* environment. MOEMS or Optical MEMS technology has recently demonstrated a strong potential in biomedical imaging applications due to its outstanding advantages including, for instance, low cost, high operating frequency and convenience of batch fabrication. 

Integrating micro-optical components with optical imaging setups for *in vitro* observation and analysis has been demonstrated by many researchers. Here, we will review several works as examples, showing the capability and potential of the miniaturized optical components. Diekmann et al. demonstrated the use of a waveguide-integrated optical chip, which hosts the biological sample and utilizes the waveguide as the illumination source, and a standard low-cost microscope to acquire super-resolved images of single molecules via two different approaches [17]. The waveguides composed of a material with high refractive-index contrast can provide a strong evanescent field that is used for single-molecule switching and fluorescence excitation, thus enabling chip-based single-molecule localization microscopy. Additionally, multimode interference patterns can induce spatial fluorescence intensity variations that enable fluctuation-based super-resolution imaging [17]. The waveguides on the chip are used for total-internal-reflection fluorescence excitation, and the detection signal are collected and recorded by conventional optical microscope. By achieving visualization of fenestrations in liver sinusoidal endothelial cells with feature size smaller than the diffraction limit, the setup shows its great potential on upgrading standard optical microscopes to super-resolution microscopic tools [17]. Yokokawa et al. reported a micro-device that is equipped with an optical fiber, a microlens and a micro-prism for in situ observation and analysis of cells [58]. The optical components were used to compose a TIR-based chip to perform TIR fluorescence microscopy on adherent cells. The cells were cultured under continuous medium perfusion in a microfluidic channel that is located on top of the TIR-based chip (Figure 8a,b). The device was evaluated by monitoring the location of insulin granules in mouse pancreatic cells. The system allowed a higher imaging signal-to-noise ratio than obtained with epifluorescence microscopy.

Advanced microelectronics and emerging computational microscopy techniques have provided another scheme to bypass various limitations of conventional optical microscopy. Typically, bioimaging is performed at the microscopic scale and this usually requires lenses integrated in a microscopic system. Conventional optical microscope can be downsized to a portable device by utilizing commercial electronic devices. Breslauer et al. developed a microscope attachment for cell-phones that is capable of both bright-field and fluorescent imaging [59]. This microscope utilizes trans-illumination configuration with standard microscope eyepieces and objectives; magnification and resolution can be adjusted using different objectives. This cellphone microscope shows promising results for clinical use by imaging *P. falciparum*-infected and sickled red blood cells in bright-field mode, and *M. tuberculosis*-infected sputum samples in fluorescent mode with light-emitting diode (LED) excitation. Furthermore, lens-free imaging has matured as a modality competitive with traditional lens-based microscopy. By placing an image sensor (CMOS/CCD) beneath a biological sample and using it to record light transmission pixel by pixel, a diffraction pattern resulting from the sample is recorded directly on the image sensor without being optically imaged or magnified by any lens elements, traditional lens components in a microscopic system can be abandoned [60]. Such lens-free (or lensless) scheme shows great flexibility on the integration format, and offers great possibilities to provide a compact and non-diffraction-limited imaging technique with large field-of-view, of the same size as the sensor size and a numerical aperture close to 1, since the large-area detector is placed very close to the sample. In the last decade, multiple lens-free on-chip microscope modus have been proposed to perform imaging on microorganisms [61] and cells [62] etc., showing exciting breakthroughs in point-of-care applications. Yang’s group demonstrated an optofluidic microscope (OFM) for lens-free contact imaging on *Caenorhabditis elegans* [61]. The OFM mobilizes the specimen along a microfluidic channel by laminar flow, and the channel is positioned directly over the image sensor. A tilted array of metallic apertures is patterned directly over the image sensor. Each aperture is carefully positioned at the center of a pixel of the image sensor, so that shadows of the specimens can be sampled by these sub-micron apertures as they flow along the microfluidic channel. By using OFM, automated phenotype characterization of different *Caenorhabditis elegans* mutant strains, as well as imaging of spores and single cellular entities, have been demonstrated [61]. The same group later on presented the implementation of color OFM prototypes color imaging of red blood cells infected with *Plasmodium falciparum*, a particularly harmful type of malaria parasites and one of the major causes of death in the developing world [62]. Ozcan’s research group has developed different lens-free holographic imaging platforms for on-chip cytometry and diagnostics, with either fluorescent imaging format [63] or shadow imaging format [64]. Zhu et al. demonstrated a compact platform that integrated imaging cytometry and fluorescent microscopy and could be attached to a cell phone [65]. The resulting device could be used to rapidly image bodily fluids for cell counts or cell analysis (Figure 8c). Wei et al. proposed using gold and silver nanobeads as specific labels to identify and count the number of CD4 and CD8 cells in a cell suspension [66]. CD4 and CD8 cells are specific types of T lymphocytes, and their relative populations are important for evaluating the stage of human immunodeficiency virus (HIV) infection or acquired immune deficiency syndrome (AIDS), as well as for evaluating the efficacy of antiretroviral treatment. Counting the relative populations of these cells can be challenging, however, because the only significant difference between these cells is in the types of proteins expressed on their membranes. Under a conventional optical microscope, both types of cells look virtually identical. However, by using gold nanoparticles functionalized with anti-CD4 antibodies to label CD4 cells and silver nanoparticles functionalized with anti-CD8 to label CD8 cells, the different spectral response of the labelled cells can be used to discriminate these two types of cells with greater than 95% accuracy using a machine learning algorithm.

### 4.2. Micro-optical Components for in vivo Bioimaging

In vitro imaging of biological samples remains as one of the main tools for diagnosis of disease. However, it typically requires invasive sample collection (e.g., biopsy) steps as well as time consuming sample preparation protocols (e.g., fixation and staining) [67]. On the other hand, in vivo optical imaging techniques enable real-time visualization and might provide accurate information for diagnostics, making them highly desirable for diagnostics. In this section, we will briefly review two main in vivo optical imaging techniques, i.e., microendoscopy and OCT, that hold various integration format with the micro-optical components.

Microendoscopy is a promising approach for *in vivo* imaging. It combines conventional intravital microscopy and miniature endoscopy, using a narrow-diameter optical probe that provides minimally invasive access to internal organs that are otherwise difficult to reach with conventional instruments [68]. Optical microendoscopy provides spatial resolution that can approach that of a conventional water-immersion objective lens [69]; is compatible for use with multiple contrast modalities including epifluorescence, two-photon excited fluorescence, and second-harmonic generation; and has been used in both live animals and humans [70]. Kim et al. developed an *in vivo* confocal and multiphoton microendoscopic imaging system which integrated with gradient index lenses [68]. The cellular resolution of the system was demonstrated by imaging the micrometer-scale structures of the fluorescence labelled pollen grains. Besides, the boundaries of spherical starch granules in a sliced fresh potato were acquired with the second-harmonic generation imaging mode. Individual dendritic cells in the epidermis and dermis, blood vessels, as well as collagen fibrillar structure in live mice, were clearly imaged [68]. By using high-end micro-optical structures, such as high-resolution microlenses [69], the performance of the imaging system can be further improved. Barretto et al. demonstrated two-photon imaging of dendritic spines on hippocampal neurons and dual-color nonlinear optical imaging of neuromuscular junctions in live mice [69]. 

Many advances in microendoscopy have been reported in the last decade. Most endoscope designs involve three key components: optical fibers used for efficient laser pulse delivery and signal collection, micro-optics used for imaging, such as microlenses, and miniaturized scanning devices, such as micromirrors [71]. Rogers et al. demonstrated an integrated microendoscope and presented images of fixed biological samples acquired by the microendoscope to demonstrate its ability to image the cellular structure of tissue [72]. To be useful as an endoscopic device, the instrument should be no more than a few millimeters in diameter. As shown in Figure 9, the objective includes a 1-mm plano-spherical glass lens mounted in a custom precision micromount and three aspheric microlenses printed via grayscale lithography in hybrid sol-gel glass. The device incorporates a MEMS scanning grating, and the grating actuator is an electrostatic comb-drive actuator designed to scan the grating in resonance. The illumination is provided by a light-emitting diode (LED) that is coupled to a multimode fiber. The light from the fiber is collected by a 2-mm plano-spherical glass condenser lens mounted in a precision micromount. The device has a 250-µm-diameter field of view (FOV) and a working distance of 300 µm. The working distance allows for optical sectioning of epithelial cells below the tissue surface [72]. To verify the imaging ability of this optical system in relevant biological samples, such as cervix, oral epithelial cells and tissues were imaged by the system due to their structural and biomolecular similarity between the oral cavity and cervix. Human oral carcinoma cells were labelled and imaged, cell membranes were clearly visible and the results were comparable with a Zeiss optical microscope [72].

OCT is a 3D microscopic imaging technique that is especially suitable for *in vivo* imaging applications, such as biomedical tissue cross-sectional analysis. With an axial resolution of less than 15 μm and cross-sectional imaging to a depth of ~1–3 mm, it has significant potential in ophthalmology, cardiology, gastroenterology and oncology applications, etc. OCT is an interferometric imaging technique that employs light sources (typically in the near-infrared) with relatively large spectral bandwidths (e.g., ~10–300 nm), to achieve cross-sectional biomedical imaging. OCT systems in general employ a fiber-based Michelson interferometer to measure the backscattered light from an object. This back-scattered light is used to calculate the reflectivity or scattering potential profile of the biological sample along the probe beam direction. By scanning the probe beam in the transverse direction, 3D images of the object can be reconstructed [67]. With the increasing availability of micro-optical components, OCT becomes a promising non-invasive imaging technique that lends itself to a compact, relatively cost- effective and portable architecture for diagnostic imaging applications. Aljasem et al. presented a membrane-based microfluidic tunable microlens and an electrostatic 2D scanning micromirror which were fabricated using silicon and polymer-based MEMS technologies [73]. These components were assembled inside a 4.5 mm diameter probe of an endoscopic OCT system for beam focus and steering, and the system was used to test a slide of wood with a surface layer of varnish and a leek skin [73]. Mu et al. demonstrated a prototype of an OCT bioimaging endoscopic probe utilizing a MEMS micromirror as the light beam manipulator [74]. To exert a large mirror platform tilt angle, electrothermal bimorph actuator was selected as the micromirror structure for its large deflection and low driving voltage, of which the latter is of particular importance for clinical applications. The MEMS micromirror and the silicon optical bench (SiOB) assembly were enclosed within a biocompatible, transparent and waterproof polycarbonate tube equipped with toroidal-lens for in vivo diagnostic applications. The muscle and skin next to the hind leg of a 6-week-old male mouse was tested by the OCT probe, epimysium, perimysium and endomysium layers were observed, and even the blood vessels that enclosed within the underneath layer of the epimysium can be easily distinguished from perimysium. Similarly, the superficial epidermis of the mouse leg skin tissue and the dermis layer were also successfully detected [74].

## 5. Conclusions

We have briefly introduced basic features of several imaging techniques that permit harvesting spatiotemporal information from living organisms, such as CT, MRI, etc. In particular, we have pointed out that optical microscopy plays a predominant role in observing tissues, cells and biomolecules. Over the last decade, a fast-growing interest was notice for developing micro-optical components and miniaturized optical imaging tools, these optical microdevices have shown great potential on imaging living organisms, tissues, cells, as well as subcellular structures, both *in vivo* and *in vitro*. We introduced several typical micro-optical components and their fabrication technologies, followed by the integration of these components with instruments or devices for in vivo and in vitro bioimaging, respectively. 

We hope this review has made clear that micro-optical components can advantageously contribute to the bioimaging applications and the integration of such components with microscopic instruments is key to the improvement of the performance of an optical imaging system and to the miniaturization of the system. Aiming to bring effective medical diagnostic tools to patients, the continued development of optical bioimaging instruments that are cost-effective, sensitive and accurate is essential. In this quest, the advances in optical microdevices/microsystems can transform the field of biomedical optics, therefore playing an increasing important role. Conventional optical imaging techniques, such as optical microscope, endoscopes and OCT systems can be redesigned to provide highly integrated and miniaturized imaging tools, as described in this review. It is probable that the miniaturization of the imaging instruments can compromise their imaging performances, such as low spatial and temporal resolution. Therefore, further developments have to be performed so as to improve the performance of these tools up to, or even beyond, the level of their respective gold standards. We believe that the technical eruptions in the aspect of micro-optics will play a prominent role in the new bioimaging techniques development.

## Figures and Tables

**Figure 1 micromachines-10-00405-f001:**
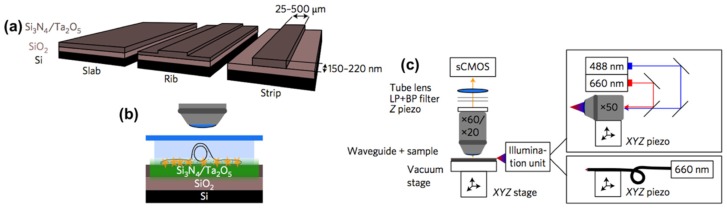
Schematic of the waveguide platform: (**a**) Channel-like waveguide geometries are realized by etching the SiO_2_ slab waveguide either partially or completely; (**b**) Light guided inside the waveguide is the source of the evanescent field illuminating samples on top of the surface; (**c**) The optical set-up. (Reproduced with permission [17], Copyright 2017, Nature Publishing Group).

**Figure 2 micromachines-10-00405-f002:**
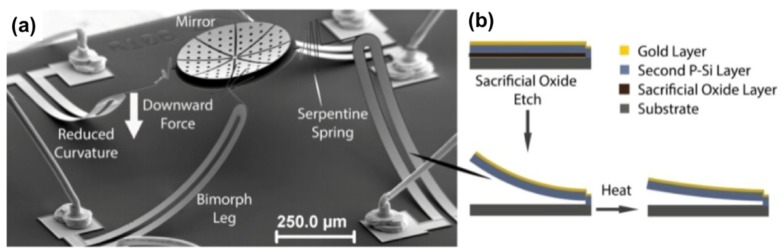
Pictures of a microelectromechanical systems (MEMS) micromirror using electrothermal actuation: (**a**) Scanning electron microscope (SEM) image of the micromirror device; (**b**) Illustration of bimorph layers before and after oxide etch and reduced curvature due to heating. (Reproduced with permission [24], Copyright 2015, Optical Society of America).

**Figure 3 micromachines-10-00405-f003:**
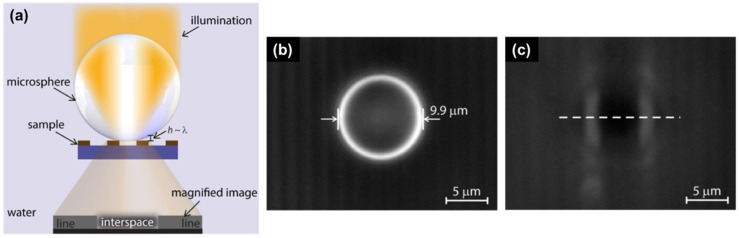
Microsphere lens used for super-resolution imaging: (**a**) A microsphere is positioned on a grating structure and illuminated from the front, the light reflected by the grating allows detecting a magnified image. When the distance *h* between the microsphere and the grating is small enough (of order of the illumination wavelength *λ*), the near-field evanescent wave carrying the fine details of the grating can become propagating in the high refractive index sphere, and later in the medium where it is to be collected by the microscope objective; (**b**,**c**) Optical microscopy images obtained by positioning a 9.9 μm microsphere on the grating. The image of (**b**) is focused on the microsphere’s center plane, and the corresponding image (**c**) is focused on the image plane. (Reproduced with permission [35], Copyright 2016, American Chemical Society).

**Figure 4 micromachines-10-00405-f004:**
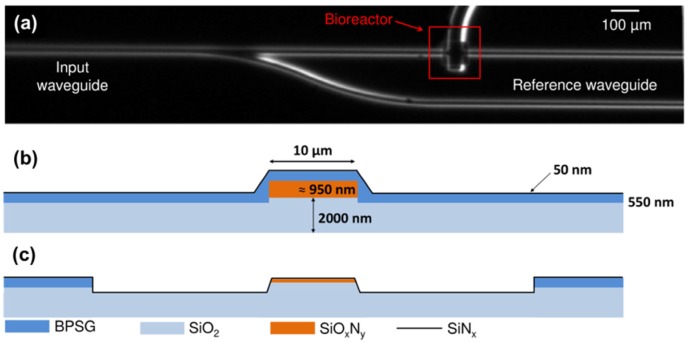
Fabrication procedure of silicon oxynitride waveguides: (**a**) Optical image of the top view of the waveguide; (**b**,**c**) Schematic cross-sections of the waveguide structure before (**b**) and at the bioreactor well (**c**). (Reproduced with permission [38], Copyright 2014, Institute of Physics).

**Figure 5 micromachines-10-00405-f005:**
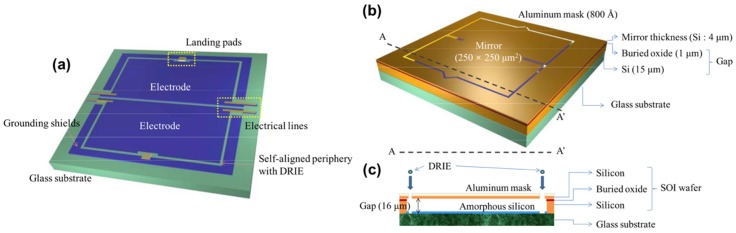
Schematic of a micromirror: (**a**) The electrodes on the glass substrate; (**b**) The micromirror structure; (**c**) Cross-section view of the device and its deep reactive-ion etching (DRIE) process. (Reproduced with permission [44], Copyright 2009, Institute of Physics).

**Figure 6 micromachines-10-00405-f006:**
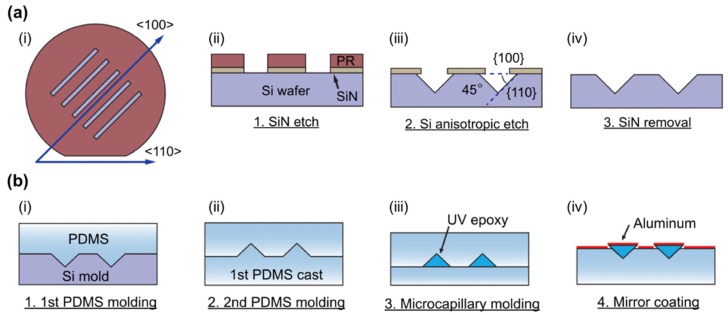
Micromirror fabrication process: (**a**) Line patterns with 45° alignment to (110) wafer flat were formed on a SiN wafer. Anisotropic Si etch leads to cross-section of right angle isosceles triangle. RIE etch of remaining SiN gives final Si master mold; (**b**) Polydimethylsiloxane (PDMS) microchannels were made by two step replica molding from the Si master mold. UV curable photopolymer was filled and cured in the micro- channels. The released structure was coated with aluminum using sputtering. (Reproduced with permission [48], Copyright 2014, American Institute of Physics).

**Figure 7 micromachines-10-00405-f007:**
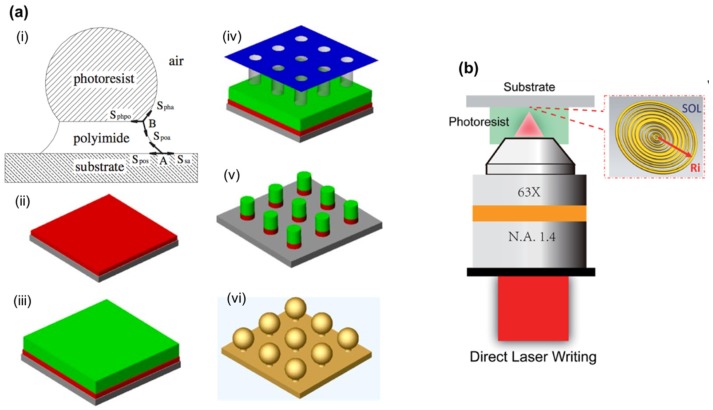
Fabrication methods for microlenses: (**a**) Illustration of a micro-ball lens formation in the thermal reflow process. A polyimide layer is coated on a Si wafer beneath a photoresist layer. Photolithography process is used to pattern these two materials through a mask. Micro-ball lens array with a pedestal can be generated after heat reflow; (**b**) Schematic diagrams of direct laser writing to form a super-oscillatory lens. ((**a**) Reproduced with permission [49], Copyright 2004, Institute of Physics; (**b**) Reproduced with permission [51], Copyright 2018, Royal Society of Chemistry).

**Figure 8 micromachines-10-00405-f008:**
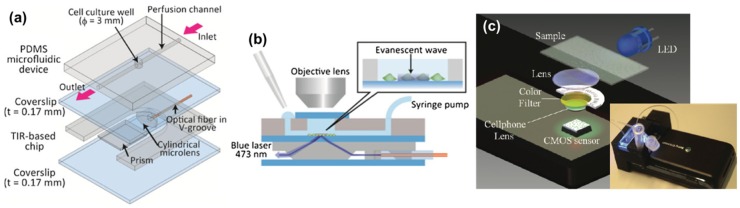
Microsystems used for in vitro imaging: (**a**) Schematic of a microdevice that is equipped with an optical fiber, a microlens and a micro-prism; (**b**) The integrated device for *in situ* observation and analysis of cells; (**c**) Schematic of an optofluidic device used for fluorescent imaging cytometry on a cellphone, the insert is a picture of the setup. ((**a**,**b**) Reproduced with permission [58], Copyright 2012, Springer; (**c**) Reproduced with permission [65], Copyright 2011, American Chemical Society).

**Figure 9 micromachines-10-00405-f009:**
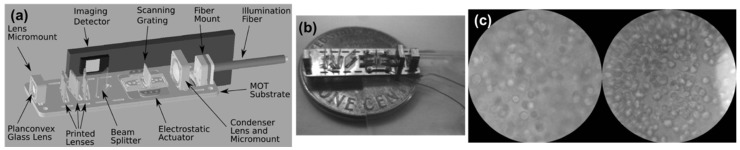
A microendoscope used for in vivo imaging: (**a**) Optical components mounted on the substrate; (**b**) The integrated device shown above a United States penny for scale; (**c**) Widefield image of cells labeled with gold nanoparticles, the image taken with the microendoscope is on the right, and a similar image taken with a Zeiss Axiovert 100 M is shown on the left for comparison. (Reproduced with permission [72], Copyright 2008, Society of Photo-Optical Instrumentation Engineers).

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
