# Peer review of "Micro-optical Components for Bioimaging on Tissues, Cells and Subcellular Structures"

_micromachines, 2019, doi:10.3390/mi10060405_

Round 1

Reviewer 1 Report

In the past two decades, micromachined optical components have been  widely used in bio-imaging applications. This review article presents some devices, but most of them, especially those devices shown in figures, are not typical. I do not think this review is a significant contribution to the field.

Author Response

We agree with the reviewer that micro-optical components have been extensively integrated in bio-imaging instruments and been widely used. Micro-waveguides, mirrors and lenses, as typical micro-optical components, plays very important role in such field. Several review papers also described the importance of these components and their applications in bio-imaging, such as Reference 15, 22, 56, 63. Our review article aims to present the progress of these components in bio-imaging applications, the typical works are chosen as the figures to show the recent innovations. We believe that the structure and the content of this article will appeal to the readers with a biological and optical background, who are active in (bio-)analytical sciences and may fully profit for their applications from the newly developed technologies and devices that are based on micro-optics.

Reviewer 2 Report

Nice review paper. Thank you!

Author Response

Thank you very much for the comment

Reviewer 3 Report

The manuscript entitled “Micro-optical Components for Bio-imaging on Tissues, Cells and Sub-cellular Structures”  aims to provide an updated review of advanced optical based microsystems and technology devoted to biological substrate investigation.  The paper is well written, and clearly and well organized with a sequence of chapters reporting from hardware microdevices to applications on in-vitro and in –vivo Biosystems. These two last parts may need some more specifications and updates, as detailed to the Authors, before the paper can be considered suitable for publication in “Micromachines”.

Specific comments to the paper:

Chapter 4 promises a review of applications of micro-optical components for bioimaging of cells, tissues and sub-cellular structures. Actually, some parts of subsequent text seems still to be too much devoted to describe the devices, and to their advantages rather than real applications

This applies for example to text at lines 398-409, or lines 437-448, which should be implemented with examples of real application on biological substrates.

At lines 410-424 – “Ref 17” should be repositioned, or repeated at the end of the respective sentences, to make clearer that it is referred to the ability of optical nanoscopy techniques to show in situ subcellular structures smaller than the diffraction limit, including  fenestrations forming sieve-plate superstructures surrounded by thicker actin bundles, with  actin colocalization with the plasma membrane in liver cells.

Also, more specific examples for applications of optofluidic based systems are to be given from the respective References cited, i.e. Ref 57: Lensless high-resolution on-chip optofluidic  …automated phenotype characterization of different Caenorhabditis elegans mutant strains besides …. Electrokinetic-Drive-Based OFM System to image single cells and spores; REF 58:  color imaging of red blood cells (RBCs) , to improve diagnosis of infection with Plasmodium falciparum.

Similarly, Point 4.2 has to be revised by implementing with actual examples of in vivo diagnostic applications. For example, demonstration of microendoscopy potential of Ref 64 ranges from vegetal pollen and  pine embryo to in vivo labelled mouse tissues. Labelled mouse tissues are also shown in vivo through fluorescence imaging with high-resolution microlenses (Ref 65).

Also subsequent text and references are recalled mainly describing the “technology“ rather then examples of real application.

A final remark regards the fact that references of this chapter are by now rather “historical”, and a further effort is required to the Author to report on most recent updates.

Author Response

The authors would like to thank the reviewers as their contributions helped us improve the quality of the manuscript. We have updated the manuscript as guided by the reviewers’ comments. For ease of tracking, following conventions are used in this file:

•           The reviewer’s comments have been written in italic and in order.

•           Author answers have been written in plain text. 

•           The parts added to the manuscript have been written in bold. 

Reviewer(s)’ Comments to Author:

Reviewer # 3

Comments:

The manuscript entitled “Micro-optical Components for Bio-imaging on Tissues, Cells and Sub-cellular Structures” aims to provide an updated review of advanced optical based microsystems and technology devoted to biological substrate investigation.  The paper is well written, and clearly and well organized with a sequence of chapters reporting from hardware microdevices to applications on in-vitro and in-vivo biosystems. These two last parts may need some more specifications and updates, as detailed to the Authors, before the paper can be considered suitable for publication in “Micromachines”.

Specific comments to the paper:

Comment 1 (C1): Chapter 4 promises a review of applications of micro-optical components for bioimaging of cells, tissues and sub-cellular structures. Actually, some parts of subsequent text seems still to be too much devoted to describe the devices, and to their advantages rather than real applications. 

Answer 1 (A1): Following the reviewer’s comment, the applications of micro-optical components were emphasized in Chapter 4, new sentences were added on line 402 – line 406:

For example, multi-photon microscopes can investigate cortical micro-architecture in animals’ brain with single cell resolution [55]. Also in regard to detecting levels of the tumor suppressing protein p53 in cells, fluorescence microscopy has been used to image cancerous tissues that were stained by specially designed oncolytic adenoviruses, the latter being programmed to replicate if the cellular p53 level is low [56,57].

And on line 444 – line 451:

Conventional optical microscope can be downsized to a portable device by utilizing commercial electronic devices. Breslauer et al. developed a microscope attachment for cell-phones that is capable of both bright-field and fluorescent imaging [59]. This microscope utilizes trans-illumination configuration with standard microscope eyepieces and objectives; magnification and resolution can be adjusted using different objectives. This cellphone microscope shows promising results for clinical use by imaging P. falciparum-infected and sickled red blood cells in bright-field mode, and M. tuberculosis-infected sputum samples in fluorescent mode with LED excitation. 

And on line 461 – line 471:

Yang’s group demonstrated an optofluidic microscope (OFM) for lens-free contact imaging on Caenorhabditis elegans[61]. The OFM mobilizes the specimen along a microfluidic channel by laminar flow, and the channel is positioned directly over the image sensor. A tilted array of metallic apertures is patterned directly over the image sensor. Each aperture is carefully positioned at the center of a pixel of the image sensor, so that shadows of the specimens can be sampled by these sub-micron apertures as they flow along the microfluidic channel. By using OFM, automated phenotype characterization of different Caenorhabditis elegansmutant strains, as well as imaging of spores and single cellular entities, have been demonstrated [61]. The same group later on presented the implementation of color OFM prototypes color imaging of red blood cells infected with Plasmodium falciparum, a particularly harmful type of malaria parasites and one of the major causes of death in the developing world [62].

And on line 501 – line 510:

Kim et al. developed an in vivoconfocal and multiphoton microendoscopic imaging system which integrated with gradient index lenses [64]. The cellular resolution of the system was demonstrated by imaging the micrometer-scale structures of the fluorescence labelled pollen grains. Besides, the boundaries of spherical starch granules in a sliced fresh potato were acquired with the second-harmonic generation imaging mode. Individual dendritic cells in the epidermis and dermis, blood vessels, as well as collagen fibrillar structure in live mice, were clearly imaged [68]. By using high-end micro-optical structures, such as high-resolution microlenses [69], the performance of the imaging system can be further improved. Barretto et al. demonstrated two-photon imaging of dendritic spines on hippocampal neurons and dual-color nonlinear optical imaging of neuromuscular junctions in live mice [69].

And on line 525 – line 529:

To verify the imaging ability of this optical system in relevant biological samples, such as cervix, oral epithelial cells and tissues were imaged by the system due to their structural and biomolecular similarity between the oral cavity and cervix. Human oral carcinoma cells were labelled and imaged, cell membranes were clearly visible and the results were comparable with a Zeiss optical microscope [72].

And on line 558 – line 563:

The muscle and skin next to the hind leg of a 6-week-old male mouse was tested by the OCT probe, epimysium, perimysium and endomysium layers were observed, and even the blood vessels that enclosed within the underneath layer of the epimysium can be easily distinguished from perimysium. Similarly, the superficial epidermis of the mouse leg skin tissue and the dermis layer were also successfully detected [74].

C2: This applies for example to text at lines 398-409, or lines 437-448, which should be implemented with examples of real application on biological substrates.

A2: Following the reviewer’s comment, the text at lines 398-409 was rewritten on line 398 – line 413 in the revised manuscript:

Recent advances in photonics and imaging techniques offer new possibilities for optical imaging systems and sensor devices for fundamental studies and future biomedical applications. Ongoing developments suggest that an impressive diversity of biological and medical questions can be answered using optical microscopy. Advanced optical techniques have been proposed for achieving high-resolution imaging of sample of interest. For example, multi-photon microscopes can investigate cortical micro-architecture in animals’ brain with single cell resolution [55]. Also in regard to detecting levels of the tumor suppressing protein p53 in cells, fluorescence microscopy has been used to image cancerous tissues that were stained by specially designed oncolytic adenoviruses, the latter being programmed to replicate if the cellular p53 level is low [56,57]. However, these techniques are frequently accompanied by the need for high-end, bulky and expensive optical set-ups that are rather cumbersome to operate and to maintain, especially for untrained users. Such drawbacks have motivated researchers to minimize and integrate optical components in imaging systems for in situ analysis and observation of bio-samples under in vitro environment. MOEMS or Optical MEMS technology has recently demonstrated a strong potential in biomedical imaging applications due to its outstanding advantages including, for instance, low cost, high operating frequency and convenience of batch fabrication. 

And the text at lines 437-448 was rewritten on line 441 – line 459 in the revised manuscript:

Advanced microelectronics and emerging computational microscopy techniques have provided another scheme to bypass various limitations of conventional optical microscopy. Typically, bio-imaging is performed at the microscopic scale and this usually requires lenses integrated in a microscopic system. Conventional optical microscope can be downsized to a portable device by utilizing commercial electronic devices. Breslauer et al. developed a microscope attachment for cell-phones that is capable of both bright-field and fluorescent imaging [59]. This microscope utilizes trans-illumination configuration with standard microscope eyepieces and objectives; magnification and resolution can be adjusted using different objectives. This cellphone microscope shows promising results for clinical use by imaging P. falciparum-infected and sickled red blood cells in bright-field mode, and M. tuberculosis-infected sputum samples in fluorescent mode with LED excitation. Furthermore, lens-free imaging has matured as a modality competitive with traditional lens-based microscopy. By placing an image sensor (CMOS/CCD) beneath a biological sample and using it to record light transmission pixel by pixel, a diffraction pattern resulting from the sample is recorded directly on the image sensor without being optically imaged or magnified by any lens elements, traditional lens components in a microscopic system can be abandoned [56]. Such lens-free (or lensless) scheme shows great flexibility on the integration format, and offers great possibilities to provide a compact and non-diffraction-limited imaging technique with large field-of-view, of the same size as the sensor size and a numerical aperture close to 1, since the large-area detector is placed very close to the sample.

C3: At lines 410-424 – “Ref 17” should be repositioned, or repeated at the end of the respective sentences, to make clearer that it is referred to the ability of optical nanoscopy techniques to show in situ subcellular structures smaller than the diffraction limit, including fenestrations forming sieve-plate superstructures surrounded by thicker actin bundles, with actin colocalization with the plasma membrane in liver cells. 

A3: Thanks to the reviewer’s comment, Reference 17 is repositioned at several places to elucidate the function of the waveguide-integrated optical chip, this part was modified on line 414 – line 428 in the revised manuscript:

Integrating micro-optical components with optical imaging setups for in vitro observation and analysis has been demonstrated by many researchers. Here, we will review several works as examples, showing the capability and potential of the miniaturized optical components. Diekmann et al. demonstrated the use of a waveguide-integrated optical chip, which hosts the biological sample and utilizes the waveguide as the illumination source, and a standard low-cost microscope to acquire super-resolved images of single molecules via two different approaches [17]. The waveguides composed of a material with high refractive-index contrast can provide a strong evanescent field that is used for single-molecule switching and fluorescence excitation, thus enabling chip-based single-molecule localization microscopy. Additionally, multimode interference patterns can induce spatial fluorescence intensity variations that enable fluctuation-based super-resolution imaging [17]. The waveguides on the chip are used for total-internal-reflection fluorescence excitation, and the detection signal are collected and recorded by conventional optical microscope. By achieving visualization of fenestrations in liver sinusoidal endothelial cells with feature size smaller than the diffraction limit, the setup shows its great potential on upgrading standard optical microscopes to super-resolution microscopic tools [17]. 

C4: Also, more specific examples for applications of optofluidic based systems are to be given from the respective References cited, i.e. Ref 57: Lensless high-resolution on-chip optofluidic…automated phenotype characterization of different Caenorhabditis elegans mutant strains besides …. 

Electrokinetic-Drive-Based OFM System to image single cells and spores; REF 58:  color imaging of red blood cells (RBCs), to improve diagnosis of infection with Plasmodium falciparum.

A4: Following the reviewer’s comment, examples for applications of optofluidic based systems were added in the revised manuscript on line 461 – line 471:

Yang’s group demonstrated an optofluidic microscope (OFM) for lens-free contact imaging on Caenorhabditis elegans[61]. The OFM mobilizes the specimen along a microfluidic channel by laminar flow, and the channel is positioned directly over the image sensor. A tilted array of metallic apertures is patterned directly over the image sensor. Each aperture is carefully positioned at the center of a pixel of the image sensor, so that shadows of the specimens can be sampled by these sub-micron apertures as they flow along the microfluidic channel. By using OFM, automated phenotype characterization of different Caenorhabditis elegansmutant strains, as well as imaging of spores and single cellular entities, have been demonstrated [61]. The same group later on presented the implementation of color OFM prototypes color imaging of red blood cells infected with Plasmodium falciparum, a particularly harmful type of malaria parasites and one of the major causes of death in the developing world [62]. 

C5: Similarly, Point 4.2 has to be revised by implementing with actual examples of in vivo diagnostic applications. For example, demonstration of microendoscopy potential of Ref 64 ranges from vegetal pollen and pine embryo to in vivo labelled mouse tissues. Labelled mouse tissues are also shown in vivo through fluorescence imaging with high-resolution microlenses (Ref 65).

A5: Following the reviewer’s comment, examples of in vivodiagnostic applications were added on line 501 – line 510:

Kim et al. developed an in vivoconfocal and multiphoton microendoscopic imaging system which integrated with gradient index lenses [68]. The cellular resolution of the system was demonstrated by imaging the micrometer-scale structures of the fluorescence labelled pollen grains. Besides, the boundaries of spherical starch granules in a sliced fresh potato were acquired with the second-harmonic generation imaging mode. Individual dendritic cells in the epidermis and dermis, blood vessels, as well as collagen fibrillar structure in live mice, were clearly imaged [68]. By using high-end micro-optical structures, such as high-resolution microlenses [69], the performance of the imaging system can be further improved. Barretto et al. demonstrated two-photon imaging of dendritic spines on hippocampal neurons and dual-color nonlinear optical imaging of neuromuscular junctions in live mice [69]. 

C6: Also subsequent text and references are recalled mainly describing the “technology” rather than examples of real application.

A6: Thanks to the reviewer’s comment, more examples of real applications on using Micro-optical components for in vivo bio-imaging were added in the revised manuscript on line 525 – line 529:

To verify the imaging ability of this optical system in relevant biological samples, such as cervix, oral epithelial cells and tissues were imaged by the system due to their structural and biomolecular similarity between the oral cavity and cervix. Human oral carcinoma cells were labelled and imaged, cell membranes were clearly visible and the results were comparable with a Zeiss optical microscope [72].

And on line 551 – line 552:

and the system was used to test a slide of wood with a surface layer of varnish and a leek skin [73].

And on line 558 – line 563:

The muscle and skin next to the hind leg of a 6-week-old male mouse was tested by the OCT probe, epimysium, perimysium and endomysium layers were observed, and even the blood vessels that enclosed within the underneath layer of the epimysium can be easily distinguished from perimysium. Similarly, the superficial epidermis of the mouse leg skin tissue and the dermis layer were also successfully detected [74].

C7: A final remark regards the fact that references of this chapter are by now rather “historical”, and a further effort is required to the Author to report on most recent updates.

A7: Following the reviewer’s comment, Reference 55, 56, 57, 59 were added in the manuscript, and several references were replaced with more recent articles.